# Spatial Gene Expression of Human Coronary Arteries Revealed the Molecular Features of Diffuse Intimal Thickening in Explanted Hearts

**DOI:** 10.3390/ijms26051949

**Published:** 2025-02-24

**Authors:** Boaz Li, Samuel Leung, Maria Elishaev, Wan Hei Cheng, Giuseppe Mocci, Johan L. M. Björkegren, Chi Lai, Amrit Singh, Ying Wang

**Affiliations:** 1Department of Pathology and Laboratory Medicine, University of British Columbia, Vancouver, BC V6T 2B5, Canada; bli1025@student.ubc.ca (B.L.); samuel.leung@hli.ubc.ca (S.L.); maria.elishaev@hli.ubc.ca (M.E.); melody.cheng@hli.ubc.ca (W.H.C.); clai39@providencehealth.bc.ca (C.L.); 2Centre for Heart Lung Innovation, University of British Columbia, Vancouver, BC V6Z 1Y6, Canada; 3Department of Medicine, Karolinska Institute, 171 77 Solna, Sweden; giuseppemocci@gmail.com (G.M.); johan.bjorkegren@ki.se (J.L.M.B.); 4Department of Genetics and Genomic Sciences, Institute of Genomics and Multiscale Biology, Icahn School of Medicine at Mount Sinai, New York, NY 10029, USA; 5Division of Anatomical Pathology, Providence Health Care, St. Paul’s Hospital, Vancouver, BC V6Z 1Y6, Canada; 6Department of Anesthesiology, Pharmacology and Therapeutics, University of British Columbia, Vancouver, BC V6T 2B5, Canada

**Keywords:** diffuse intimal thickening, spatial gene expression, smooth muscle cells, RNA quality control

## Abstract

Diffuse intimal thickening (DIT) is a pre-clinical stage of atherosclerosis characterized by thickened intima. The molecular basis of its susceptibility to atherogenesis is unknown, and mechanistic investigations cannot be performed in commonly used mouse models, in which DIT does not exist. Vascular smooth muscle cells (SMCs) are the predominant cell type that occupies the intima and media of DIT. The molecular differences between these two layers may reveal the earliest phenotypic changes in SMCs to promote atherosclerosis. We benchmarked the RNA quality of human coronary arteries from autopsies (n = 7) and explanted hearts (n = 7) and performed Visium spatial gene expression on tissue sections with DIT. Although autopsy samples met the RNA quality standard for Visium (DV200 ≥ 30%), only arteries from explanted hearts exhibited reliable sequencing performance. Genes enriched in TGF-β-mediated remodeling of the extracellular matrix were overrepresented in the intima. SMCs enriched in the intima are dedifferentiated, but unlike those in the atherosclerotic lesions, they are not pro-inflammatory. Our findings indicate that autopsy samples are not ideal to distinguish subtle differences among cell phenotypes. SMCs in thickened intima may lead to lipid retention but not necessarily the onset of atherosclerosis.

## 1. Introduction

The intimal layer of aortas in commonly used mouse models of atherosclerosis contains endothelial cells lying on top of the internal elastic lamina. In contrast, humans exhibit diffuse intimal thickening (DIT) before atherosclerosis develops, especially in atherosclerosis-prone arteries such as the coronary arteries. In DIT, the intimal layer thickens with enriched smooth muscle cells (SMCs) and extracellular matrix [1]. DIT is universal in both sexes, and progression is seen with increasing age [2]. DIT was observed as early as 36 weeks of gestation and these regions are prone to atherosclerosis development in later life [3]. The American Heart Association defines DIT as a normal pre-clinical stage with well-organized intimal SMCs in the absence of lipid deposition or robust infiltration of immune cells [4].

Why the intimal layer of DIT regions is particularly prone to atherogenesis remains largely unknown. The intimal and medial layers of DIT regions are both enriched with SMCs, which are a major source of proteoglycans and collagens, forming the extracellular matrix in the vessel wall. One of these proteoglycans enriched in the intima of DIT, biglycan, has a high binding capacity for low-density lipoprotein [5], which could promote the retention of extracellular lipids and SMC-foam cells during early atherogenesis [6]. Intimal SMCs can undergo phenotypic changes and adopt various phenotypes in mouse and human atherosclerotic lesions [7]. Human SMCs in the intima of DIT express fewer myosin heavy chains than the fully differentiated contractile SMCs in the medial layer, suggesting a dedifferentiated phenotype [1]. Our previous studies using single-cell RNA-sequencing have found a unique dedifferentiated SMC population in mouse atherosclerotic lesions, which expresses stem cell marker Sca1 [8,9]. Pseudotime trajectory analysis suggested that Sca1^+^ SMCs are in transition from contractile to pro-inflammatory phenotypes [8,10], playing a key role in the onset of atherosclerosis in mouse models. It is tempting to assume that similar stem-like SMCs exist in the intima of DIT in humans, priming these regions for atherogenesis upon exposure to risk factors such as cholesterol and inflammation. However, the Sca1 gene is not present in humans. Therefore, a thorough molecular profile of SMCs in the DIT may reveal their earliest phenotypic changes and risk factors that are essential for atherogenesis.

Knowledge of the molecular features of DIT is limited, as previous studies relied on microscopic observation and immunostaining of coronary arteries from autopsies. Determining the molecular features in a specific region requires high-throughput RNA sequencing in the spatial context. Spatial transcriptomics of coronary arteries would allow for the gene expression of intimal and medial layers to be observed separately, which is especially useful for comparing the molecular features between intimal and medial SMCs in DIT. However, only one successful application in human coronary arteries has been reported so far [11], suggesting that spatial transcriptomics is still at its infant stage in answering clinically relevant questions about coronary artery disease [12]. Of note, human coronary arteries from autopsies are easily accessible compared to those collected from explanted hearts. RNA degradation in autopsy samples may introduce bias in the analysis and further complicate data interpretation. Before more studies pursue spatial transcriptomics to understand the etiology of coronary artery disease, the potential pitfall of using autopsy samples needs to be investigated.

In the present study, we applied the spatial gene expression technology Visium to obtain whole transcriptomic profiles of the intimal and medial layers of coronary arteries from formalin-fixed paraffin-embedded (FFPE) tissue sections. Visium uses DV200 (i.e., the percentage of fragments >200 nucleotides) to select FFPE samples of high RNA integrity, and the minimum requirement is 30% [13]. Data quality between autopsies and explanted hearts was compared. Coronary arteries from heart transplant patients were selected to reveal the molecular features of thickened intima, which cannot be studied using commonly used mouse models or single-cell RNA sequencing of dissociated blood vessels.

## 2. Results

Movat staining showed that the intima is thicker than the media (Figure 1A) in DIT regions. SMCs expressing contractile protein marker MYH11 were enriched in both intima and media, whereas CD45^+^ leukocytes were rarely observed (Figure 1A and Appendix A), confirming that the selected DIT regions met the pathology definition. When the layout of Visium SD capture spots was overlaid on top of the confocal images, it was estimated that RNA extracted from one capture spot (Figure 1A, white circle) could originate from more than one nucleus (blue) and from different vessel layers (white arrow).

The characteristics of sample donors are summarized in Appendix A. No significant differences in RNA fragmentation, in terms of DV200 value, were observed between autopsy and explanted heart samples (Figure 1B, left panel). However, the yield of RNA from autopsy samples was roughly half of that from the explanted heart samples, even though the autopsy samples were stored in FFPE blocks for less than 2 years, suggesting RNA degradation due to post-mortem events (Figure 1B, right panel). For DIT samples with a DV200 above 30%, autopsy samples had a much higher background signal from genomic DNA, as indicated by the estimated portion of UMI counts from genomic DNA (Figure 1C). The mitochondrial gene transcripts were enriched in the autopsy samples, especially in the regions of intima and media (Figure 1D). One sample, autopsy sample A3, had only 8.4% of transcripts mapped to probes and failed sequencing quality control (Figure 1C). This sample was ruled out from further analysis.

At an FDR of 1%, there were 336 genes more enriched in the intimal and medial layers of DIT from explanted hearts (Figure 1E). Gene ontology analysis (Figure 1F) showed that these genes encode proteins that are involved in cellular components like “Focal Adhesion” and “Cell-Substrate Junction” (e.g., *ITGB1*, *JAK2*, *ACTG1*, *VIM, ACTB*), “Vesicle” (*ANXA1, UBB, ACTB, MYO1D*), and “Cell-Cell Junction” (e.g., *CCN2*, *ASPN*, *SERPINE1, ELN*) (Appendix A). The top over-represented genes in autopsy samples are olfactory receptors (e.g., *OR8D1, OR6C2, OR6T1, OR2B3, OR2S2, OR6C75*) (Figure 1E).

Capture spots at the apical layer of the intima could be a mix of endothelial cells (ECs) and SMCs (yellow spots, Figure 2B, panel a). This was confirmed by the expression of both ECs marker *PECAM1* and SMCs marker *MYH11* (Figure 2B, panel b). After these capture spots were ruled out, the remaining spots located in the subendothelial part of the intima (blue spots, intima-subendothelium) (Figure 2B, panel c) were compared with those in the media (red spots) (Figure 2B, panel e). Both the blue and red capture spots were enriched with SMCs expressing MYH11 but not with ECs expressing PECAM1 (Figure 2B, panel d and f), suggesting that they contain predominantly intimal and medial SMCs, respectively. Differential gene expression analysis (Figure 2C) revealed that the intimal SMC-dominated blue capture spots were enriched with genes in the pathways of TGF-β regulation of extracellular matrix (e.g., *VCAN, LTBP2*), syndecan-1 pathway and collagen biosynthesis (e.g., *COL8A1, COL16A1*), extracellular matrix organization (e.g., *MMP2*), the RAGE pathway (e.g., *TNFRSF11B*), PDGF genes and receptors (e.g., *PDGFRA*), and glycosaminoglycan metabolism (e.g., *VCAN, BGN*), as compared to the red capture spots, which were occupied mainly by medial SMCs (Figure 2D, Appendix A). SMCs in the media were enriched with genes for muscle contraction (e.g., *ACTG2, ACTC1, MYH11*), suggesting a contractile phenotype. Interestingly, complement components *C3* and *C7* in the alternative complement pathway were overrepresented in the media.

Some signature genes of Sca1^+^ SMCs, including *VCAN, BGN, MMP2, TNFRSF11B, COL8A1, EFEMP1,* and *LTBP2*, were more enriched in the intimal SMC-enriched blue capture spots as compared to medial SMC-enriched red capture spots (Figure 2E, panel a, and Appendix A). However, these intima SMC-dominated capture spots did not turn on genes in the inflammatory response (e.g., *VCAM1, LUM, DCN, SPP1, C3, C4B, CXCL12*, and *SERPINA3*), as Sca1^+^ SMCs did (Appendix A). DEGs in intimal SMC-enriched capture spots overlapped the most with signature genes of sub-cluster 5 from human carotid lesions (Appendix A). They are involved in the pathway of TGF-β regulation of extracellular matrix (e.g., *VCAN, EFEMP1*) (Figure 2E, panel b). However, these intimal SMC-enriched capture spots lack the osteogenic and inflammatory features (e.g., *LUM, DCN, VCAM1*, C3, C7) presented in dedifferentiated SMC clusters 6–8 in human carotid lesions (Figure 2E, panel b, Appendix A). Capture spots in the media were enriched with *MYH11, CNN1, ACTA2, ACTG2*, and *LMOD1*, sharing similar features with contractile SMCs in both human and mouse atherosclerotic lesions (Figure 2E).

## 3. Discussion

Spatial gene expression technology is a rapidly developing technique for obtaining transcriptome profiles in the tissue context. Compared to single-cell RNA sequencing, archived FFPE tissue blocks can be analyzed. The application of spatial gene expression technology on human specimens has thus far focused on tumor biopsies, which are easily accessible from live patients. To study vascular biology, particularly coronary arteries, post-mortem hearts are easily accessible, widely used, and well-accepted for pathology studies. The discovery of DIT and its relationship with atherosclerosis are the results of autopsy studies [3]. However, in the era of next-generation ‘omics’, the use of post-mortem tissues for RNA sequencing is debatable.

Our study is the first to benchmark RNA quality and spatial sequencing performance of post-mortem coronary arteries. Forensic studies have indicated that post-mortem RNA degradation is tissue-specific and gene-specific [14]. RNA was found to be relatively more stable in the brain and cardiac muscle than in the aorta [14]. Sequencing quality assessment metrics, including estimated UMI from genomic DNA and reads mapped to probes, are also tissue-specific [15]. Therefore, coronary arteries from heart transplant patients are needed as a control to assess RNA quality prior to post-mortem degradation. Our results have shown that even if autopsy samples have met the minimal requirement of the DV200 RNA integrity test [13], this test does not accurately indicate RNA degradation in autopsy samples. It only predicts that the length of RNA fragments is suitable for library construction [16]. The estimated UMI from genomic DNA and mitochondrial transcripts is high in some autopsy samples because DNA and mitochondrial RNA are less sensitive to post-mortem degradation than mRNA [17]. Therefore, we do not recommend solely relying on DV200 for quality control. Measure of RNA yield per tissue section could be an additional criterion to rule out samples with degraded RNA before proceeding to sequencing. If only post-mortem blood vessels are available, samples with both high RNA yield and DV200 should be prioritized.

Despite RNA degradation, the transcriptome profiles of autopsy tissues may still be valuable in identifying gross differences between diseased and healthy populations. When the clinical characteristics of donors are considered, heart failure of explanted heart samples is a cofounding factor for data interpretation. The overrepresented olfactory receptors in autopsy samples (donors without clinically diagnosed heart failure) align with impaired olfactory function and dysregulated expression in the hearts of heart failure patients [18]. However, to distinguish subtle differences between phenotypes (e.g., SMCs in the intima and media), our results have shown that RNA degradation could jeopardize the resolution of spatial gene expression. Hence, it is important to critically assess the value of post-mortem tissues based on the research questions.

Due to limited access to human samples and in vitro models, it is still unknown how DIT is formed. One theory is that DIT is an adaptive response to mechanical stresses [4]. Well-preserved mRNA from explanted hearts highlights the heterogeneity of SMC-enriched capture spots, suggesting that intimal SMCs are phenotypically different from medial SMCs. One limitation of the Visium SD platform is that it does not have single-cell resolution to directly identify capture spots with a pure SMC population. We assumed that the MYH11^+^ capture spots in the subendothelial and the medial layers contain predominantly SMCs because CD45 staining is rarely seen in DIT, ruling out the possibility of leukocytes contributing to the transcriptome profiles of these capture spots. Although pericytes and fibroblasts can express SMC contractile markers, pericytes are distributed in the microvessels [19], and fibroblasts are in the adventitia of non-atherosclerotic coronary arteries [20]. We used gene ontology analysis to predict the functional status of intimal SMCs as the outcome of stimuli in the microenvironment of the intima. It suggested that SMCs in the intima may have been exposed to TGF-β, an important cytokine that stimulates the production of extracellular matrix by SMCs, including versican (encoded by *VCAN*) and biglycan (encoded by *BGN)* [21,22]. In the current study, gene expression analysis indicated that *VCAN* and *BGN* were enriched in the subendothelial regions of the intima, which is consistent with their protein expression in DIT [23] and at least part of the phenotypic outcome of TGF-β activation. Increased shear stress generated by the blood flow stimulates the secretion of active TGF-β from ECs [24]. Together with platelet-derived growth factor, TGF-β attracts SMCs to establish and enlarge arteries during vascular development [21]. However, we do not think that shear stress-induced secretion of TGF-β from ECs can simply explain DIT formation. Rodent aortas face much greater shear stress than humans [25] but do not have DIT. The molecular features of DIT seen in this study (from patients over 40 years old) are the results of decades of tissue remodeling, and aging itself can contribute to DIT formation after birth. Increased *MMP2* in the intima has also been observed in the thickened intima of older monkeys as compared to the thin intima of the young, both without atherosclerosis [26]. *MMP2* can activate TGF-β production by SMCs in vitro, increasing their expression of collagen and proteoglycans [27]. Coronary arteries from young donors would be a better fit to investigate how DIT was originally formed during embryonic development. DIT in heart failure patients, especially those from patients with ischemic cardiomyopathy, is of particular interest because they remain at a pre-clinical stage when other parts of the coronary arteries that are exposed to the same genetic and circulating risk factors have developed lesions. We can infer the proatherogenic features of SMCs and risk factors in the lesion microenvironment by comparing SMCs in these DITs to those from the atherosclerotic lesions.

Intimal SMCs are hypothesized to derive from migration and clonal expansion of a subset of medial SMCs, like Sca1^+^ SMCs in mouse atherosclerotic lesions [28,29]. Without cell lineage tracing tools and specific phenotypic markers for intimal SMCs, we cannot directly test the origin, stimulus of phenotypic transition, and function of intimal SMCs in humans. Nevertheless, our results suggested that Sca1^+^ SMCs from mouse lesions reflect a phenotype in response to inflammation in the lesions, and human intimal SMCs in the DIT represent the phenotypic outcome of no or low inflammation activation. *Cellular response to cytokine stimulus* and *cytokine-mediated signaling pathway* are the top 2 gene ontology features of Sca1^+^ SMCs from the lesions of *Ldlr^−^*^/*−*^*Apob^100^*^/*100*^ mice [9], which were not found in human SMCs in the intima of DIT. To predict the functional differences between intimal and medial SMCs in the DIT, interpretation based on individual DEGs is a traditional challenge [30]. For example, *VCAN* and *FBLN1* increased the proliferation and migration of SMCs in vitro [31,32], but they were enriched in the intima and media, respectively, in the current study. The relationship between gene ontology pathways and cellular function of Sca1^+^ SMCs from *Apoe^−/−^* mouse lesions has been tested: activation of the complement cascade led to the secretion of complement component C3, which promotes proliferation and inflammation [8]. Since activation of the complement cascade is absent in intimal SMCs, we do not expect that intimal SMCs in DIT will be hyperproliferative and pro-inflammatory as Sca1^+^ SMCs. Moreover, trajectory analysis of sub-clusters from human carotid lesions indicated that sub-cluster 5, sharing the most featured markers with intimal SMCs in the DIT, is distinct from other dedifferentiated SMCs that were considered as chondrogenic, osteogenic, or pro-inflammatory [9]. All of the evidence above supports the theory of Nakashima et al.: intimal SMCs in DIT are in a stable status, and they are not adequate to drive disease progression [6].

Intimal SMCs in DIT could be in a primed state that can readily transdifferentiate to different phenotypes and inflammation can be the key stimulus for them to switch on more proatherogenic features. Such a group of primed SMCs was found in healthy mouse aortas, and the authors suggested that inflammation drives the initial phenotypic changes in atherogenesis [33]. Proteoglycans in the extracellular matrix play a significant role in modulating cell response to inflammation. *LUM* and *DCN*, highly expressed by the osteogenic and inflammatory SMCs in advanced human carotid lesions [9], modulate the bioactivity of cytokines, contributing to inflammation and calcification of atherosclerotic lesions [34,35,36]. In the current study, the lack of these “proatherogenic” proteoglycans in SMCs from DIT regions suggests that the interplay between SMC phenotypes and the extracellular matrix is important for disease progression. Intimal SMCs may pave the way for foam cell formation by secreting *BGN* [28], but the formation of SMC-foam cells will not necessarily activate cell death or chronic inflammation within human lesions [37]. Studying how inflammation affects SMC phenotype and its role in tissue remodeling will help explain why, while everyone has DIT, those with chronic inflammatory disorders have a higher risk of developing coronary artery disease, independent of traditional risk factors for cardiovascular disease [38].

## 4. Materials and Methods

### 4.1. Human Samples

Human coronary lesions were obtained from autopsies or explanted hearts donated by heart transplant patients. Autopsy samples were collected 12–72 h after the donors were deceased. Explanted samples were collected during the surgeries and fixed within 6 h. For both types of samples, coronary arteries were fixed in 10% buffered formalin (CAT#16004-128, VWR, West Chester, PA, USA) at room temperature for 24 h. The fixed tissues were then embedded in paraffin using a Leica Microsystems ASP 6025 processor (Leica, Wetzlar, Germany), and the FFPE blocks were stored at the Bruce McManus Biobank (St. Paul’s Hospital, University of British Columbia). Using the modified American Heart Association classification, a pathologist assessed coronary arteries after Movat pentachrome staining. Regions of DIT or pathologic intimal thickening (PIT) were selected [39] from heart transplant patients (n = 7) and autopsies (n = 7). DIT is featured by an intima thicker than the media. The intima is rich in SMCs and extracellular matrix, with little or no lipid accumulation. PIT has foam cells and a lipid pool of extracellular lipids.

### 4.2. Immunofluorescence Staining and Confocal Microscopy

To distinguish DIT and PIT, we stained the FFPE tissue sections with SMC lineage marker myosin heavy chain 11 (MYH11, CAT#ab82541, 1:100) (Abcam, Cambridge, UK) and pan-leukocyte marker CD45 (CAT#368502, 1:100) (Biolegend, San Diego, CA, USA). Briefly, 5 µm FFPE tissue sections were dewaxed, rehydrated, and boiled in 10 mM sodium citrate, 0.05% Tween buffer (pH 6.0) for 30 min using an autoclave for antigen retrieval. After blocking with 10% goat serum for 60 min, sections were incubated with the primary antibodies at 4 °C overnight, washed, and then incubated with Alexa Fluor secondary antibodies (Invitrogen CAT#A11008 and #A21236, 1:200) at room temperature for 1 h, and imaged under a Zeiss LSM880 inverted confocal microscope.

### 4.3. RNA Extraction and Quality Tests

Eight successive 10 µm thick FFPE sections were collected into RNase-free tubes immediately after sectioning, and RNA was extracted following the manufacturer’s protocol (CAT#73504, Qiagen, Hilden, Germany). Total RNA was condensed by lyophilization. RNA concentration was measured by a Qubit Fluorometer to calculate the yield of RNA from each tissue section. RNA quality was determined by DV200 using an RNA 6000 Pico kit (Agilent, CAT# 5067-1513) on a Bioanalyzer. DIT samples with a DV200 ≥ 30% were sectioned, mounted on Visium SD slides (10× Genomics, Pleasanton, CA, USA), and stained by H&E. Library preparation was performed according to the manufacturer’s protocol and sequenced at a minimum of 25,000 reads per capture spot.

### 4.4. Spatial Gene Expression Analysis

Visium data were processed using the Seurat package (v5.0.1) [40]. To assess the RNA quality of each sample, spatial heatmaps were used to visualize the proportion of mitochondrial genes within each capture spot. Regularized negative binomial regression was used to adjust for cellular sequencing depth [41]. Principal component analysis (PCA) was performed on normalized data, and the top 50 principal components were retained. Harmony (v1.2.1) [42] was used to correct for batch effects. Unsupervised clustering was performed using Harmony-corrected components using the Louvain algorithm. Clustered capture spots were visualized with the Uniform Manifold Approximation and Projections (UMAP) algorithm, using the first 30 Harmony-corrected components. To view the distribution of clusters in their spatial context, clustered capture spots were overlaid on top of H&E images for each tissue section. For differential gene expression analysis, specific capture spots within our regions of interest were selected using location tags collected interactively using the SpatialDimPlot function from the Seurat R-library.

### 4.5. Differential Gene Expression Analysis

We identified capture spots from the intimal and medial layers of the blood vessels (from the apical side to the external elastic lamina) by overlaying the Visium capture spots on the corresponding H&E image. We first fit a linear mixed model with LIMMA-VOOM (v3.50.3) [43] to determine differentially expressed genes (DEGs) of the coronary arteries (intimal and medial layers combined) between autopsy and explanted heart samples using pseudo-bulk gene counts. Then, for coronary arteries from the explanted heart samples only, we used LIMMA-VOOM to determine DEGs between the intima (below the endothelium to the internal elastic lamina) and media (below the internal elastic lamina to the external elastic lamina), adjusting for patient-specific random effects. A false discovery rate of 0.01 was used to correct for multiple testing. DEGs were ranked by log2-fold change and visualized by volcano plots using ggplot2 (v3.5.1). DEGs were assessed for gene ontology (Cellular Component 2023) or pathways enrichment (BioPlanet 2019) through the web-based platform Enrichr [44].

### 4.6. Comparison to SMCs in Single-Cell RNA Sequencing Studies

To compare the similarity of SMCs in DIT and in atherosclerotic lesions, DEGs in either the intimal or medial SMC-enriched capture spots were mapped against the top 100 specific markers of Sca1+ SMCs from atherosclerotic lesions of *Apoe−/−* mice [8], *Ldlr−/−Apob100/100* mice, and the 8 SMC sub-clusters found in human carotid atherosclerotic lesions [9] to find overlapped genes. Detailed analysis methods to obtain specific markers have been described previously [8,9]. The average counts of representative signature genes in the intimal and medial SMC-enriched capture spots were visualized by a heatmap using ggplot2.

### 4.7. Statistical Analysis

For statistical analyses performed using GraphPad Prism (version 5.0, GraphPad Software, Boston, MA, USA), the normality of distribution was assessed using the D’Agostino and Pearson normality tests (α = 0.05). Normally distributed data were presented as mean ± SEM, and non-normally distributed data were presented as median and interquartile range. Normally distributed data with equal variances were analyzed using unpaired *t*-tests. Data that did not meet parametric assumptions were analyzed using the Wilcoxon test. Statistical significance was defined as *p*-value < 0.05.

## 5. Conclusions

Post-mortem coronary arteries, which are frequently collected by biobanks, have degraded RNA, which is not suitable for identifying subtle differences in the transcriptome profiles. Coronary arteries from explanted hearts allow for a more faithful representation of spatial gene expression across the vessel wall as compared to ones from autopsy hearts. Thickened intima in diffuse intimal thickening, which exists in everyone, is unlikely to undergo atherogenesis without additional stimuli such as inflammation.

## Figures and Tables

**Figure 1 ijms-26-01949-f001:**
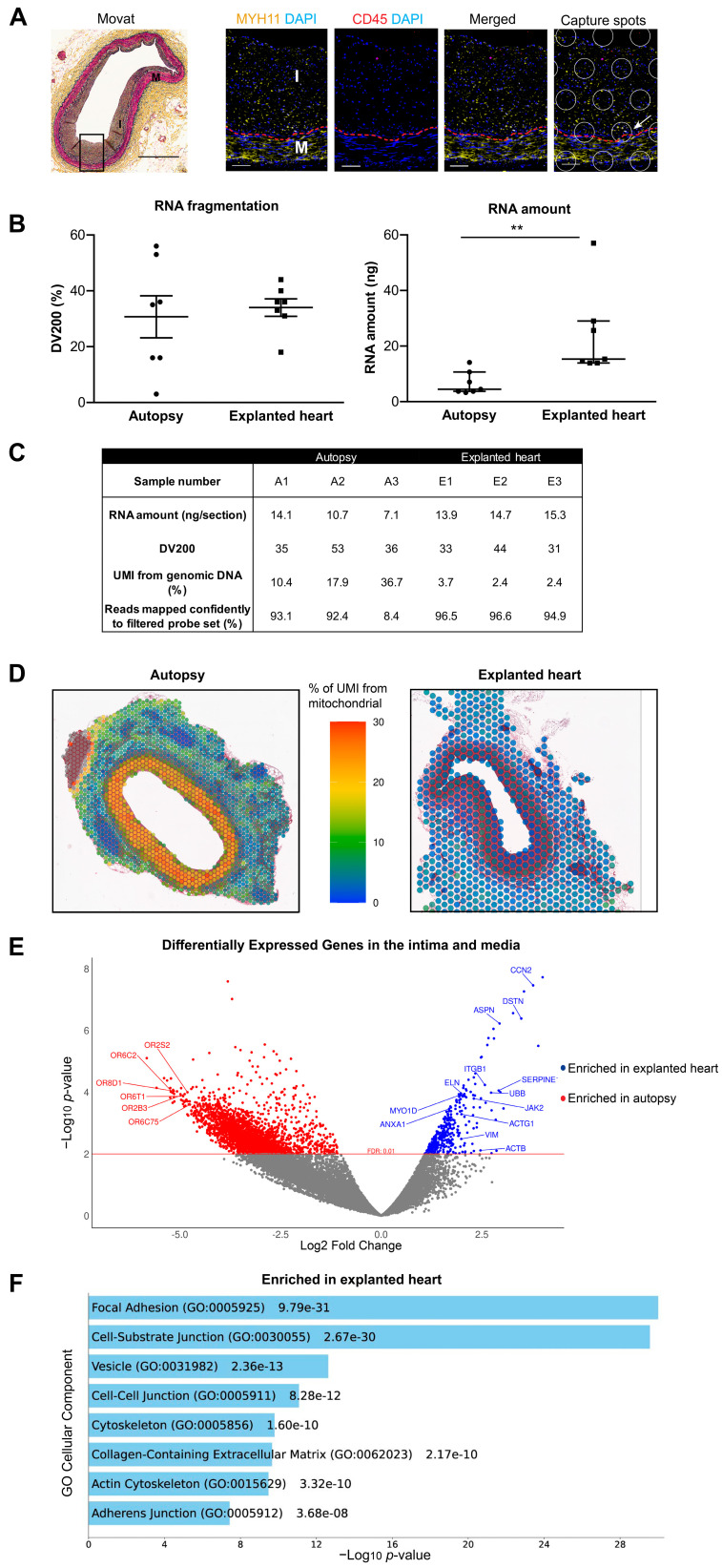
Comparison of RNA quality and transcriptome profiles of coronary arteries from autopsy and explanted heart. (**A**) Movat staining of a human coronary artery with DIT (black square, scale bar = 700 µm). Confocal imaging of this DIT region with MYH11 (yellow), CD45 (red), and nuclei (blue) (scale bar = 50 µm). White circles indicate the layout of Visium SD capture spots. Red dashed line: internal elastic lamina; I: intima; M: media. Arrow points to a capture spot that contains cells from both intima and media. (**B**) Assessment of DV200 (left, mean ± SEM, unpaired *t*-test, *p* = 0.694) and amount (right, median with interquartile range, Wilcoxon test, *p* = 0.002) of RNA extracted from FFPE tissue sections of coronary arteries with DIT or PIT (n = 7 per group, ** *p* < 0.01). (**C**) Pre- and post-sequencing quality assessment of DIT samples with a DV200 ≥30% (n = 3 per group). (**D**) In situ distribution of mitochondrial gene counts in DIT regions. (**E**) DEGs enriched in the intima and media of DIT samples from autopsies (red) and explanted hearts (blue, n = 3 per group). Grey dots represent genes below the cut-off threshold of FDR = 0.01. (**F**) Gene ontology analysis (Cellular Component 2023) of DEGs enriched in the intima and media of coronary arteries with DIT from explanted heart (n = 3).

**Figure 2 ijms-26-01949-f002:**
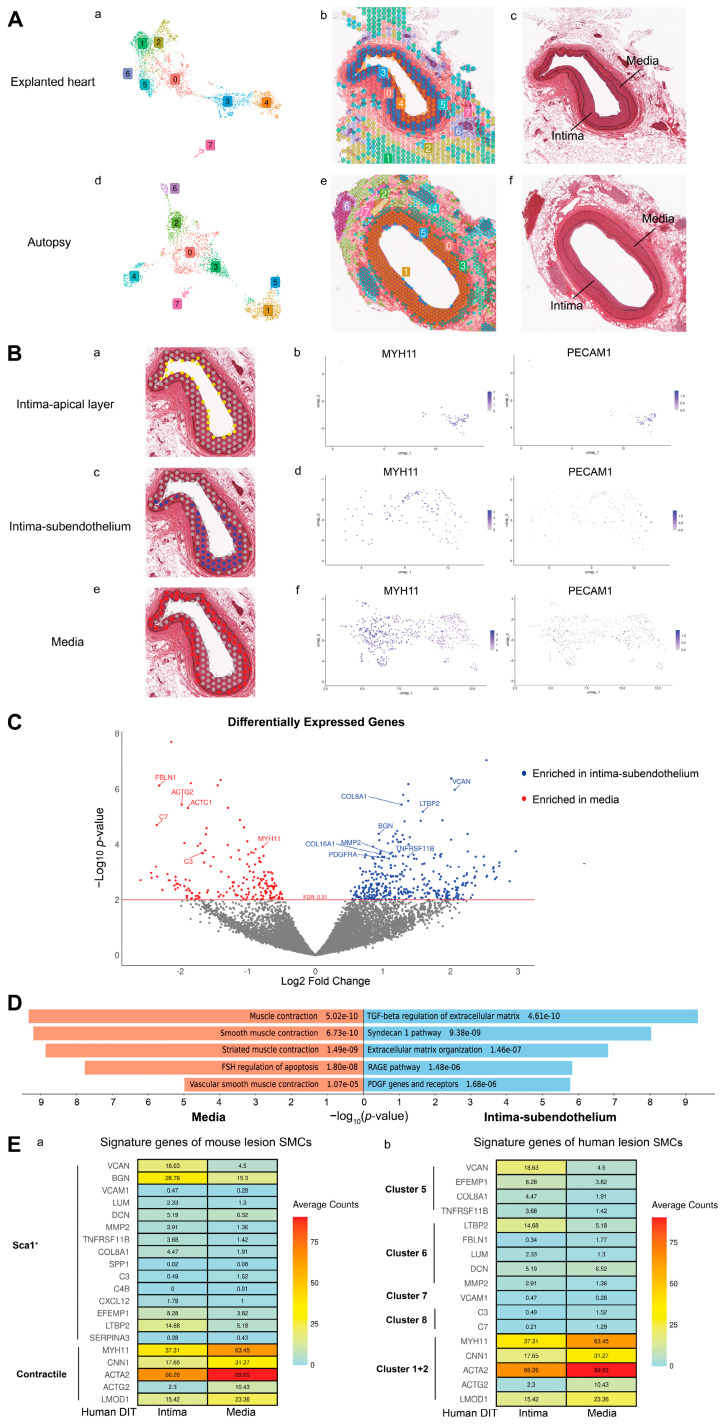
Transcriptome profiles of SMC-enriched capture spots in the DIT. (**A**) Capture spots grouped by unsupervised clustering of their transcriptome profiles (**Aa**,**Ad**), n = 3 per group) and localized in the DIT tissue sections (**Ab**,**Ae**) stained by H&E (**Ac**,**Af**). Black line: internal and external elastic lamina. Colors and numbers label different clusters of capture spots in explanted heart (a and b) and autopsy (d and e). (**B**) Expression of *MYH11* and *PECAM1* (**Bb**,**Bd**,**Bf**) in capture spots located in the apical layer ((**Ba**), yellow), subendothelium of the intima ((**Bc**), blue), and the medial layer ((**Be**), red) of DIT coronary arteries from explanted hearts (n = 3). (**C**) DEGs enriched in the blue and red capture spots in B. (n = 3) and (**D**), their gene ontology analysis (BioPlanet Pathway 2019)**.** Grey dots represent genes below the cut-off threshold of FDR = 0.01. (**E**) A heatmap showing the expression of representative genes in SMC-enriched capture spots in the human DIT (intima and media separately). Genes are top signature genes of SMCs from mouse atherosclerotic lesions (**Ea**), Sca1^+^ SMCs and contractile SMCs, and sub-clusters from human carotid lesions (**Eb**).

## Data Availability

The data that support the findings of this study appear in the main text or the Appendix A. Raw data of spatial gene expression are available from NCBI GSE283269 at https://www.ncbi.nlm.nih.gov/geo/query/acc.cgi?acc=GSE283269 (accessed on 11 February 2025). Code for data analysis and data visualization can be found at https://github.com/CompBio-Lab/VisiumVisualizations/tree/main.

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
