# Peer review of "Spatial Gene Expression of Human Coronary Arteries Revealed the Molecular Features of Diffuse Intimal Thickening in Explanted Hearts"

_ijms, 2025, doi:10.3390/ijms26051949_

Round 1

Reviewer 1 Report

Comments and Suggestions for Authors

In the present study the authors compared Visium spatial gene expression of coronary arteries obtained from autopsy and freshly explanted hearts. The study is interesting and well performed. There are only minor concerns:

1. The authors need to provide dilutions of primary and secondary antibodies used for immunofluorescence (on page 3).

Because the study is well performed there are only minor concerns. The authors stated that it is possible to study RNA from autopsy tissues by using next-generation omics technology. RNA was extracted from formalin-fixed paraffin-embedded (FFPE) tissue samples. In addition, the authors stated that FFPE RNA can be used to study gene expression and correlate disease state with tissue morphology. However, I reviewed the manuscript one more time and here are some additional comments: 1. The authors indicated that freshly isolated coronary arteries from heart transplant patients were used in the present study (page 3, first paragraph). However, supplemental Table 1 is showing that the storage time was between 1 and 19 years. Please clarify. 2. The authors need to add a statement in the section "material and methods" regarding storage of the samples in formalin-fixed paraffin-embedded (FFPE), for example: a) What is the exact composition of formalin used for fixing of the samples (provide company and catalog #). b) How long were the samples stored in formalin prior to embedding in paraffin? c) It seems that the FFPE protocol is not standardized throughout the study period. d) Though it is great for preserving the tissue structures, it can lead to RNA degradation. RNA is often degraded and chemically modified, making it challenging to analyze. How it is possible that RNA degradation is much lower in coronary arteries from obtained from explanted hearts as compared to those from autopsy despite very long storage (up to 19 years)? e) Would it be possible to compare to controls meaning that RNA is extracted directly from fresh tissues (without FFPE)? 3. The authors need to deposit the raw and processed data of RNA sequence studies to the Gene Expression Omnibus (GEO) database and provide the access # in the manuscript.

Author Response

The authors need to provide dilutions of primary and secondary antibodies used for immunofluorescence (on page 3).

This information has now been added on page 10 “Immunofluorescence staining and confocal microscopy”.

Because the study is well performed there are only minor concerns. The authors stated that it is possible to study RNA from autopsy tissues by using next-generation omics technology. RNA was extracted from formalin-fixed paraffin-embedded (FFPE) tissue samples. In addition, the authors stated that FFPE RNA can be used to study gene expression and correlate disease states with tissue morphology. However, I reviewed the manuscript one more time, and here are some additional comments:

  1. The authors indicated that freshly isolated coronary arteries from heart transplant patients were used in the present study (page 3, first paragraph). However, supplemental Table 1 is showing that the storage time was between 1 and 19 years. Please clarify.

We apologize for the confusion raised by our wording. “Fresh coronary arteries” refers to samples quickly isolated from (live) heart transplant patients and biobanked immediately. The autopsy samples typically stay in deceased donors for at least 12 hours before being taken out for biobanking (to have their family consent for donation). Compared to samples from heart transplant patients, post-mortem autolysis is unavoidable in autopsy samples. A more accurate word to replace “fresh coronary arteries” should be “freshly collected”. For explanted and autopsy tissues, the standard biobanking procedure is the same and the FFPE blocks are stored for future research. “Storage time” in Table 1 refers to the storage time of FFPE blocks, not the time in formalin. Since this is a retrospective study, we used FFPE samples already available at the biobank instead of prospectively collecting samples on an ongoing basis. Therefore, the storage time of FFPE blocks is a range. 

To clarify, we have removed the phrase “fresh” throughout the manuscript and changed “Storage time” to “FFPE blocks storage time” in supplemental Table 1. We have also added detailed descriptions in “material and methods” to emphasize the common and different procedures of biobanking explanted and autopsy samples.

  1. The authors need to add a statement in the section "material and methods" regarding storage of the samples in formalin-fixed paraffin-embedded (FFPE), for example: a) What is the exact composition of formalin used for fixing of the samples (provide company and catalog #). b) How long were the samples stored in formalin prior to embedding in paraffin? c) It seems that the FFPE protocol is not standardized throughout the study period.

The FFPE protocol is standardized for all samples. After collection, samples were fixed in 10% buffered formalin for 24 h and then they were embedded and stored at the biobank. From sample collection to storage of FFPE blocks, the only system variables that we cannot control are the time between death and sample collection for autopsies. This is because the donors’ families need to consent after the unexpected death of the donors. We cannot collect the samples before the consent is being made. For the explanted samples, heart transplant patients consent to donate their diseased hearts before the surgeries and we fixed the explanted samples within 6 h after they were taken out from the patients. We have added the details to “material and methods”:

“Autopsy samples were collected 12-72 h after the donors were deceased. Explanted samples were collected during the surgeries and fixed within 6 h. For both types of samples, coronary arteries were fixed in 10% buffered formalin (VWR, CAT#16004-128) at room temperature for 24 h. The fixed tissues were then embedded in paraffin using a Leica Microsystems ASP 6025 processor and the FFPE blocks were stored at the Bruce McManus Biobank (St. Paul’s Hospital, University of British Columbia).”

  1. d) Though it is great for preserving the tissue structures, it can lead to RNA degradation. RNA is often degraded and chemically modified, making it challenging to analyze. How it is possible that RNA degradation is much lower in coronary arteries from obtained from explanted hearts as compared to those from autopsy despite very long storage (up to 19 years)? e) Would it be possible to compare to controls meaning that RNA is extracted directly from fresh tissues (without FFPE)?

We agree that fresh tissues are ideal for RNA-sequencing purposes. However, most biobanks follow the clinical logistics to collect human samples, and patient diagnosis (pathology report) is prioritized before research. FFPE is the most widely used format for pathology diagnosis and all the biobanks. That’s why more technology development is now developing protocols for FFPE blocks. One of the aims of our study is to assess the RNA quality control standard for spatial transcriptomics using FFPE blocks: DV200 value. The RNA degradation in autopsy samples is mostly due to delayed biobanking. Tissues remain in deceased bodies and are exposed to post-mortem autolysis for at least 12 h in our case. Autolysis will break down the RNA. The explanted samples are taken from live patients during heart transplants and immediately fixed and biobanked. Fixation crosslinks enzymes needed for autolysis. Therefore, the impact of autolysis is minimal in explanted samples. This explains why their total RNA amount is much higher. The DV200 value reflects RNA fragmentation that is partially attributed to the crosslinking of RNA during fixation. This part of the degradation applies to both autopsy and explanted samples. Our results indicated that autolysis contributes to RNA degradation more than the FFPE procedure. Long-term storage could contribute to RNA degradation. For spatial transcriptomics, it is recommended to exclude tissue sections on the surface of the blocks, which have been exposed to the air. Of note, none of our sequenced samples have been sectioned open before. This explains why the RNA is still well-preserved in sealed FFPE blocks after 19 years. Some studies have benchmarked RNA quality between FFPE samples and frozen samples using animal tissues (e.g., PMID: 31093665). Longer fixation time in formalin and the FFPE format is associated with low RNA yield compared to frozen samples. Due to the nature of the retrospective study, we do not have access to fresh human tissues for this work.

  1. The authors need to deposit the raw and processed data of RNA sequence studies to the Gene Expression Omnibus (GEO) database and provide the access # in the manuscript.

This has been uploaded as mentioned in the “Data Availability Statement” (page 12). We have refreshed the status and added the link for public access.

“Data Availability Statement: The data that support the findings of this study appear in the main text or the Supplementary Material. Raw data of spatial gene expression are available from NCBI GSE283269 at https://www.ncbi.nlm.nih.gov/geo/query/acc.cgi?acc=GSE283269. Code for data analysis and data visualization can be found at https://github.com/CompBio-Lab/VisiumVisualizations/tree/main.”

Reviewer 2 Report

Comments and Suggestions for Authors

  • in the results section, the authors should more clearly differentiate the characteristics of DIT and PIT
  • in the discussions section I would like to see an in depth comparative analysis of the results obtained by the authors with other studies in the area. As it is, it is too fixed on the results, not their integration in a wider context.
  • the text lining is variable
  • for statistical analysis - the authors decribed the way are presented normally and non-normally distributed data, but in the results section the information seems to be lacking. I would like to see the results of the statistical analyses that were performed.
  • conclusion: postmortem coronary arteries should be replaced with Coronary arteries obtained after death or during autopsy

Author Response

We appreciate the reviewer's feedback. Please see our response and requested data attached.
